# Calibrated surrogate losses for robust classification with a reject option

## Abstract

The reject option allows learning models to evaluate their confidence in each prediction and to abstain from labeling inputs when the confidence in the predicted label is too weak. In this paper, we study the reject option in the presence of adversarially perturbed inputs, providing a framework for reliable and robust decision-making. A central challenge is to identify surrogate losses that are properly calibrated with respect to the adversarial reject option loss. We provide a detailed analysis of this problem and give a complete characterization of calibration for important hypothesis sets, such as generalized linear models. In contrast to standard adversarial settings, we prove that no quasiconcave loss is calibrated for these hypothesis sets. Motivated by this negative result, we introduce alternative non-quasiconcave surrogates.

## 1 Introduction and Motivation

Modern machine learning systems are increasingly deployed in safety-critical environments such as healthcare, finance, and autonomous driving, where reliability under malicious perturbations is essential. Yet, small but carefully crafted input perturbations known as adversarial examples can drastically change a model's prediction, exposing its vulnerability [6, 9, 12]. Understanding how to design robust learning algorithms under such perturbations has thus become a central question in modern machine learning. A standard approach formulates adversarial robustness as a minimax optimization problem involving the adversarial 0–1 loss, which measures classification performance against the worst-case perturbation. However, it is well known that optimizing this adversarial loss is NP-hard for most hypothesis classes. Consequently, most algorithms rely on surrogate losses, whose optimization is tractable. A central question is whether the minimizers of these surrogate losses are also exact or approximate minimizers of the original adversarial loss. Addressing this question has led to the introduction of key notions such as consistency and calibration which were originally considered in the setting of standard classification[3, 11]. Basically, Consistency ensures that minimization of the true risk associated with surrogate loss should lead to the minimization of the true risk associated with the adversarial 0-1 loss, while calibration is often considered as a necessary first step toward establishing consistency. It was shown in [2], that unlike in standard classification [3], convex losses fail to be calibrated under adversarial perturbations. They introduced the class of quasi-concave losses and derived necessary and sufficient conditions for calibration—conditions, but restricted to linear hypothesis sets. Building on this, [1] extended the study to generalized linear models and one-layer neural networks, while also analyzing consistency, which, unlike in standard classification, does not necessarily follow from calibration in the adversarial setting. More recently, [10] extended the framework to adversarial learning with a reject option, where models are allowed to abstain from uncertain predictions [5, 4, 7]. The reject option mechanism provides an additional layer of reliability, particularly important for robust decision-making in high-risk applications. However, the analysis presented in [10] is restricted to linear classifiers, remains theoretically incomplete, and relies on calibration conditions that are difficult to verify in practice. Moreover, several key results

are conjectural, supported primarily by empirical evidence rather than rigorous proofs, and the study does not address consistency.

This work extends previous analyses by establishing calibration results for generalized linear classifiers in the adversarial reject-option setting. Our findings provide a complete characterization of calibrated surrogate losses and show that quasi-concave losses fundamentally fail to satisfy calibration in this context.

## 2 Problem setup and preliminary results

### 2.1 Problem setup

Let $\mathcal{X}$ be the instance space and $\mathcal{Y} = \{-1, 1\}$ the label space. We consider $P$ an unknown probability distribution over $\mathcal{X} \times \mathcal{Y}$. We define a classifier as a function $f : \mathcal{X} \to \mathbb{R}$, and its generalization error $\mathbb{E}[\ell_{01}(f, x, y)]$ where $\ell_{01}(f, x, y) = \mathbb{1}_{yf(x) \leq 0}$. In statistical learning, we have access to a sample $S = (X_1, Y_1), \dots, (X_n, Y_n)$ and the objective is to minimize the generalization error over a hypothesis class $\mathcal{H} = \{x \mapsto f(x, w) \mid w \in \mathbb{W}\}$, where $\mathbb{W}$ is a parameter space. More generally, we define a loss $\ell$ as a function $\ell : \mathcal{H} \times \mathcal{X} \times \mathcal{Y} \to \mathbb{R}_{\geq 0}$ and the corresponding generalization error is $\mathbb{E}[\ell(f, x, y)]$.

We consider the learning rejection framework [5], where the learner has the option to abstain from making a prediction when it is uncertain about the label. In such cases, the learner returns a symbol $\perp$ and incurs a rejection cost $c \in (0, \frac{1}{2})$.

To incorporate the rejection option in our setting, we introduce a parameter $\rho$ that allows us to determine which points should be rejected by a classifier $f$. Specifically, If $|f(x)| \leq \rho$, the instance $x$ is rejected and no label is predicted, otherwise, the prediction $\text{sign}(f(x))$ is made. For a given instance $(x, y)$, we define the rejection loss as:

$$\ell_\rho(f, x, y) = (1 - c)\mathbb{1}_{yf(x) < -\rho} + c\mathbb{1}_{yf(x) \leq \rho}. \tag{1}$$

Thus, the expected rejection generalization error is:

$$R_{\ell_\rho}(f) = \mathop{\mathbb{E}}_{(x,y) \sim P}[\ell_\rho(f, x, y)]. \tag{2}$$

We now consider the setting where the inputs are adversarially perturbed. In this paper, we assume the instance space is the $L_2$ unit ball $\mathcal{X} = B_2(0, 1)$. For each $x \in \mathcal{X}$, the pertubation adversarial set is defined as the $\ell$ ball $B_2(x, \gamma) = \{x' \in \mathcal{X}, \parallel x - x' \parallel \leq \gamma\}$ where $\gamma > 0$ is the adversarial budget. We thus defined the *adversarial reject option loss*

$$\ell_\rho^\gamma(f, x, y) = \sup_{x' : \|x - x'\| \leq \gamma} \ell_\rho(f, x', y)$$

which is the worst rejection loss incurred over an adversarial pertubation of $x \in \mathcal{X}$ within a ball of a certain radius in a norm. Importantly, the rejection loss (1) is in the form $\ell_\rho(f, x, y) = \phi(yf(x))$ where $\phi(t) = (1 - c)\mathbb{1}_{t < -\rho} + c\mathbb{1}_{t \leq \rho}$. We can easily observe that $\phi$ is non-increasing, and following derivation by [13], we get:

$$\ell_\rho^\gamma(f, x, y) = \phi\left(\inf_{x' : \|x - x'\| \leq \gamma} yf(x)\right)$$

The corresponding generalization error is thus

$$R_{\ell_\rho^\gamma}(f) = \mathop{\mathbb{E}}_{(x,y) \sim P}[\ell_\rho^\gamma(f, x, y)] = \mathop{\mathbb{E}}_{(x,y) \sim P}\left[\phi\left(\inf_{x' : \|x - x'\| \leq \gamma} yf(x)\right)\right] \tag{3}$$

Let $R^\star_{\mathcal{H}, \ell_\rho^\gamma} = \inf_{f \in \mathcal{H}} R_{\ell_\rho^\gamma}$ be called the rejection Bayes $(\ell_\rho^\gamma, \mathcal{H})$-risk. The equation (3) above establishes the adversarial reject option loss risk. However, directly minimizing this risk is computationally challenging due to the supremum (or infimum) over adversarial pertubations and the discontinuity introduced by the rejection loss (1). Therefore, practical learning algorithms typically rely on minimizing a surrogate loss $\ell$ that is easy to optimize. For a given surrogate loss $\ell$, we define the $\ell$-generalization error as

$$R_\ell(f) = \mathop{\mathbb{E}}_{(x,y) \sim P}[\ell(f, x, y)] \tag{4}$$

and the corresponding Bayes $(\ell, \mathcal{H})-$risk is defined as $R^\star_{\mathcal{H},\ell} = \inf_{f \in \mathcal{H}} R_\ell(f)$. A central question is then: under what conditions does minimizing $R_\ell(f)$ guarantee the minimization of the true adversarial reject option risk $R_{\ell_\rho^\gamma}$? To address this question, a well-known concept introduced by [11] establishes the connection between minimizing the surrogate risk (4) and minimizing the adversarial reject risk (3). This concept is referred to as $\mathcal{H}-$consistency.

**Definition 2.1.** $\mathcal{H}-$*consistency*

*Given a hypothesis set $\mathcal{H}$, we say that a loss function $\ell$ is $\mathcal{H}-$consistent with respect to the target loss $\ell_\rho^\gamma$ if the following holds:*

$$R_\ell(f_n) - R^\star_{\mathcal{H},\ell} \xrightarrow[n \to +\infty]{} 0 \quad \Rightarrow \quad R_{\ell_\rho^\gamma}(f_n) - R^\star_{\mathcal{H},\ell_\rho^\gamma} \xrightarrow[n \to +\infty]{} 0 \tag{5}$$

*for all probability distributions and sequences of $\{f_n\}_{n \in \mathbb{N}} \subset \mathcal{H}$.*

A first step toward studying consistency is the notion of calibration [11] which plays a central role in characterizing when a surrogate loss can yield consistency. Introducing this notion requires defining some quantities. Let $\eta : \mathcal{X} \to [0,1]$ be defined as $\eta(x) = P(Y = 1 | X = x)$. For a loss function $\tilde{\ell}$, we define the conditional $\tilde{\ell}$-risk $C_{\tilde{\ell}}(f, x, \eta)$ as follows:

$$\forall x \in \mathcal{X}, \ \forall \eta \in [0,1], \quad C_{\tilde{\ell}}(f, x, \eta) = \eta \tilde{\ell}(f, x, 1) + (1 - \eta) \tilde{\ell}(f, x, -1) \tag{6}$$

Moreover, we define the minimal conditional risk $C^\star_{\mathcal{H},\tilde{\ell}}$ [11] and pseudo-minimal conditional risk $C^\star_{\mathcal{H},\tilde{\ell}}$ [2] are defined as:

$$C^\star_{\mathcal{H},\tilde{\ell}}(\mathbf{x}, \eta) = \inf_{f \in \mathcal{H}} C_{\tilde{\ell}}(f, \mathbf{x}, \eta) \quad \text{and} \quad C^\star_{\mathcal{H},\tilde{\ell}}(\eta) = \inf_{f \in \mathcal{H}, \mathbf{x} \in \mathcal{X}} C_{\tilde{\ell}}(f, \mathbf{x}, \eta) \tag{7}$$

**Definition 2.2.** *(Uniform $\mathcal{H}$-Calibration) [11]*

*Let $\mathcal{H}$ be a hypothesis set. A surrogate loss function $\ell$ is said to be uniformly $\mathcal{H}$-calibrated with respect to $\ell_\rho^\gamma$ if for every $\epsilon > 0$, there exists $\delta > 0$ such that for all $\eta \in [0,1]$, $f \in \mathcal{H}$, and $\mathbf{x} \in \mathcal{X}$, we have*

$$C_\ell(f, \mathbf{x}, \eta) - C^\star_{\mathcal{H},\ell}(\mathbf{x}, \eta) < \delta \implies C_{\ell_\rho^\gamma}(f, \mathbf{x}, \eta) - C^\star_{\mathcal{H},\ell_\rho^\gamma}(\mathbf{x}, \eta) < \epsilon.$$

**Definition 2.3.** *(Uniform Pseudo-$\mathcal{H}$-Calibration) [2]*

*Given a hypothesis set $\mathcal{H}$, a surrogate loss $\ell$ is said to be uniformly pseudo-$\mathcal{H}$-calibrated with respect to $\ell_\rho^\gamma$ if, for every $\epsilon > 0$, there exists $\delta > 0$ such that for all $\eta \in [0,1]$, $f \in \mathcal{H}$, and $\mathbf{x} \in \mathcal{X}$,*

$$C_\ell(f, \mathbf{x}, \eta) - C^\star_{\mathcal{H},\ell}(\eta) < \delta \implies C_{\ell_\rho^\gamma}(f, \mathbf{x}, \eta) - C^\star_{\mathcal{H},\ell_\rho^\gamma}(\eta) < \epsilon.$$

In the setting of standard classification, it was shown in [11] that, under suitable distributional assumptions, $\mathcal{H}$-uniform calibration implies consistency. In the adversarial setting, however, one must be more cautious. In particular, uniform calibration may not imply consistency without stronger distributional assumptions on the hypothesis set $\mathcal{H}$. Moreover, as pointed out in [1], pseudo-uniform calibration does not imply consistency, even for simple hypothesis set such as linear models. Finally, we notice that Definitions 2.2 and 2.3 differ only in the use of the minimal conditional risk, namely $C^\star_{\mathcal{H},\tilde{\ell}}(x, \eta)$ versus $C^\star_{\mathcal{H},\tilde{\ell}}(\eta)$ (where $\tilde{\ell} = \ell$ or $\ell_\rho^\gamma$). The latter is often more convenient in proofs, as observed in [1]. In fact, They considered hypothesis sets where the equality $C_{\tilde{\ell},\mathcal{H}}(x, \eta) = C_{\tilde{\ell},\mathcal{H}}(\eta)$ holds (when $\tilde{\ell} = \ell_\rho^\gamma$, or $\ell$). This is also the case in the present work, and we will therefore refer to definition 2.3 when we latter write $\mathcal{H}$-calibration.
We next introduce the uniform (pseudo) calibration functions, which are particularly useful for formulating conditions under which a surrogate loss is $\mathcal{H}$ calibrated.

**Definition 2.4.** *(Uniform (pseudo) Calibration functions) [11]*

*Let $\mathcal{H}$ be a hypothesis set. The uniform calibration function $\delta$ and the uniform pseudo-calibration function $\hat{\delta}$ associated with a pair of losses $(\ell_\rho^\gamma, \ell)$ are defined, for any $\epsilon > 0$, as*

$$\delta(\epsilon) = \inf_{\eta \in [0,1]} \inf_{f \in \mathcal{H}, \mathbf{x} \in \mathcal{X}} \left\{ C_\ell(f, \mathbf{x}, \eta) - C^\star_{\mathcal{H},\ell}(\mathbf{x}, \eta) \mid C_{\ell_\rho^\gamma}(f, \mathbf{x}, \eta) - C^\star_{\mathcal{H},\ell_\rho^\gamma}(\mathbf{x}, \eta) \geq \epsilon \right\},$$

$$\hat{\delta}(\epsilon) = \inf_{\eta \in [0,1]} \inf_{f \in \mathcal{H}, \mathbf{x} \in \mathcal{X}} \left\{ C_\ell(f, \mathbf{x}, \eta) - C^\star_{\mathcal{H},\ell}(\eta) \mid C_{\ell_\rho^\gamma}(f, \mathbf{x}, \eta) - C^\star_{\mathcal{H},\ell_\rho^\gamma}(\eta) \geq \epsilon \right\}.$$

**Proposition 2.5.** *[11, Lemma 2.16]*
*Let $\mathcal{H}$ be a hypothesis set. A loss $\ell$ is uniformly $\mathcal{H}$-calibrated (or uniformly pseudo-$\mathcal{H}$-calibrated) with respect to $\ell_\rho^\gamma$ if and only if its calibration function $\delta$ (resp. its uniform pseudo-calibration function $\hat{\delta}$) satisfies*

$$\delta(\epsilon) > 0 \quad (or \quad \hat{\delta}(\epsilon) > 0) \quad for\ all\ \epsilon > 0.$$

In the rest of the paper, we focus on margin-based surrogate losses, that is, losses of the form

$$\ell(f, x, y) = \phi(yf(x)), \quad \text{where } \phi : \mathbb{R} \to \mathbb{R}_{\geq 0}. \tag{8}$$

Therefore, we may use $\phi$ and $\ell$ interchangeably in the subsequent analysis.

In this work, we investigate the calibration of surrogate losses for generalized linear classifiers. More precisely, we provide a complete characterization of surrogate losses that are calibrated in the *adversarial-reject option* setting for generalized linear classifiers. While quasi-concave even losses have been shown to be calibrated in the standard adversarial setting[1], we prove that they fail to be calibrated once the reject option is introduced. This highlights a fundamental distinction between the two settings. Motivated by this negative result, we propose alternative non–quasi-concave surrogate losses as potential candidates (see appendix for illustration).

## 2.2 Preliminary results

While some calibration results have been partially established for the case of linear classifiers [10], our goal is to extend these results to the broader hypothesis class of generalized linear classifiers. Specifically, we consider

$$\mathcal{H}_g = \{f(x) = g(w.x) + b, \quad \| w \| = 1, \quad , |b| \leq G\} \tag{9}$$

where $g : \mathbb{R} \to \mathbb{R}$ is a fixed function, $G > 0$. In particular, when $g = \text{ReLU}$, with $\text{ReLU}(x) = \max(x, 0)$, we will denote the corresponding class by $\mathcal{H}_{\text{ReLU}}$.

Let us define $\underline{m} = \max_{\alpha \in [-1,1]} (g(\alpha) - g(\alpha - \gamma))$ and $\overline{m} = \min_{\alpha \in [-1,1]} (g(\alpha) - g(\alpha + \gamma))$. Given a margin-based loss (8), let $\tilde{C}_\phi(t, \eta)$ be defined as

$$\tilde{C}_\phi(t, \eta) = \eta\phi(t) + (1 - \eta)\phi(-t), \quad \forall \eta \in [0, 1], \ \forall t \in \mathbb{R}. \tag{10}$$

**Lemma 2.6.**
*Let $\mathcal{H}_g$ be the hypothesis set defined in (9). Let us assume that $g : \mathbb{R} \to \mathbb{R}$ is a non-decreasing and continuous function with $g(1 + \gamma) < G - \rho$, $g(-1 - \gamma) > \rho - G$ and $\rho \geq \frac{1}{2} (\underline{m} - \overline{m})$. Then, If a loss $\ell$ is $\mathcal{H}_g$−uniformly calibrated with respect to $\ell_\rho^\gamma$, it is also $\mathcal{H}_g$−pseudo uniformly calibrated with respect to $\ell_\rho^\gamma$.*

This lemma plays an important role in establishing our negative results: it suffices to show that a loss $\ell$ is not $\mathcal{H}_g$-pseudo-uniformly calibrated in order to conclude that it is not $\mathcal{H}_g$-uniformly calibrated. Moreover, equivalence between the two notions can occur in certain cases, for instance when $G = +\infty$ as stipulated in [1].

Next, we provide a full characterization of pseudo-uniform calibrated losses.

**Theorem 2.7.** *[Characterization of pseudo-uniform calibrated loss]*
*Let $g$ be a non-decreasing and continuous function such that $g(1+\gamma) < G-\rho$ and $g(-1-\gamma) > \rho-G$. Let us assume $\rho \geq \frac{1}{2} (\underline{m} - \overline{m})$. Let $\phi$ be a margin-based loss. Then $\phi$ is $\mathcal{H}_g$-pseudo uniformly calibrated with respect to $\ell_\rho^\gamma$ if on only if:*

$$\inf_{\alpha \in [g(-1)-G, \ \underline{m}-\rho[ \ \cup \ ]\overline{m}+\rho, \ g(1)+G]} \tilde{C}_\phi(\alpha, \eta) > \inf_{\alpha \in [g(-1)-G, \ g(1)+G]} \tilde{C}_\phi(\alpha, \eta) \quad if\ \min(1 - \eta, \eta) \geq c, \quad \textit{(rejection)}$$

$$\inf_{\alpha \in [g(-1)-G, \ \underline{m}+\rho]} \tilde{C}_\phi(\alpha, \eta) > \inf_{\alpha \in [g(-1)-G, \ g(1)+G]} \tilde{C}_\phi(\alpha, \eta) \quad if\ \eta > 1 - c, \quad \textit{(positive classification)}$$

$$\inf_{\alpha \in [\overline{m}-\rho, g(1)+G]} \tilde{C}_\phi(\alpha, \eta) > \inf_{\alpha \in [g(-1)-G, \ g(1)+G]} \tilde{C}_\phi(\alpha, \eta) \quad if\ \eta < c, \quad \textit{(negative classification)}$$

**Corollary 2.8.** *[Necessary and sufficient condition for $\mathcal{H}_{ReLU}$-pseudo calibration]*
*Let assume that $1 + \gamma + \rho < G$ and $\rho > \gamma$. Let $\phi$ be a margin-based loss. Then a margin-based loss*
*$\phi$ is $\mathcal{H}_{ReLU}$-pseudo uniformly calibrated with respect to $\ell_\rho^\gamma$ if and only if:*

$$\inf_{\alpha \in [-G,\ \gamma-\rho[\ \cup\ ]\rho-\gamma,\ G+1]} \tilde{C}_\phi(\alpha, \eta) > \inf_{\alpha \in [-G,\ G+1]} \tilde{C}_\phi(\alpha, \eta) \quad \text{if } \min(1-\eta, \eta) \geq c\ , \quad \text{(rejection)}$$

$$\inf_{\alpha \in [-G,\ \gamma+\rho]} \tilde{C}_\phi(\alpha, \eta) > \inf_{\alpha \in [-G,\ G+1]} \tilde{C}_\phi(\alpha, \eta) \quad \text{if } \eta > 1 - c\ , \quad \text{(positive classification)}$$

$$\inf_{\alpha \in [-\rho-\gamma, G+1]} \tilde{C}_\phi(\alpha, \eta) > \inf_{\alpha \in [-G,\ G+1]} \tilde{C}_\phi(\alpha, \eta) \quad \text{if } \eta < c\ , \quad \text{(negative classification)}$$

The above result calls for several comments. In particular, for the case $\eta = \frac{1}{2}$, the calibration condition in the standard adversarial setting requires that the conditional risk attain its minimum strictly *outside* a ball centered at the origin [1]. In contrast, once the reject option is introduced, this requirement is reversed: calibration condition requires that the minimum lie *inside* such a ball. Intuitively, when $\eta = \frac{1}{2}$, rejection is preferable to making an uncertain prediction, and the calibration condition therefore enforces a stronger penalty on predictions falling outside the robust region. Moreover, calibration conditions require a jump requirement: when rejection is optimal, that is $\min(\eta, 1-\eta) \geq c$, the minimizer of the conditional risk is required to lie with the band $[\gamma - \rho, \rho - \gamma]$; once the class probabilities moves outside $[c, 1-c]$ prediction becomes optimal and the minimizer must "jump" beyond this band, namely to $[\rho + \gamma, G]$ for positive prediction or $[-G, -\rho - \gamma]$ for negative prediction.

In standard adversarial learning (without a reject option), quasi-concave losses have been shown to satisfy desirable calibration properties [2, 1]. However, when incorporating a reject option, this relationship no longer holds, and quasi-concavity may in fact prevent $\mathcal{H}_g-$uniform calibration in our setting. This motivates the following result, which states that no quasi-concave loss is $\mathcal{H}_g-$calibrated in our framework.

**Definition 2.9.** *[2][Quasi-concave even]*
*A margin-based loss $\phi$ is said to be quasi-concave even, if $\phi(\alpha) + \phi(-\alpha)$ is quasi-concave.*

**Theorem 2.10.**
*Let $g$ be a non-decreasing and continuous function such that $g(1+\gamma) < G-\rho$ and $g(-1-\gamma) > \rho-G$.*
*Let us assume $\rho \geq \frac{1}{2}\left(\underline{m} - \overline{m}\right)$. Let $\phi$ a margin-based loss be continuous, and quasi-concave even.*
*Then $\phi$ is not $\mathcal{H}_g$ uniformly calibrated.*

# 3 Conclusion and Ongoing work

In this work, we made initial progress toward understanding calibration in the adversarial reject-option setting. We provided a complete characterization of calibrated surrogate losses for generalized linear classifiers and showed that quasi-concave losses, although calibrated in the standard adversarial setting, fail to be calibrated once the reject option is introduced. Motivated by the negative result, our ongoing work explores alternative non-quasi-concave surrogate losses as potential candidates (see appendix for more details). Unlike previous studies, which relied mainly on conjectures supported by empirical evidence, we aim to provide rigorous theoretical proofs that these surrogates satisfy the required calibration conditions.

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

## A Appendix

This appendix provides detailed proofs of the theoretical results stated in the main part of the paper. For clarity, we begin by restating the keys notations definitions used throughout the paper. In Section A.2, we provide the proof of Lemma 2.6. Section A.3 is devoted to the proof of Theorem 2.7, where we characterize the necessary and sufficient conditions for pseudo-calibrationof margin-based surrogate losses. In Section A.4 we prove Corollary 2.8. In Section A.5, we show that quasi-concave even losses fail to be $\mathcal{H}_g-$calibrated in the adversarial reject option setting.

### A.1 Preliminary Notation

### A.1 Preliminary Notation and Assumptions

Let $\mathcal{X} \subseteq \mathbb{R}^d$ be the instance space and $\mathcal{Y} = \{-1, 1\}$ the label space. We denote by $P$ an unknown probability distribution over $\mathcal{X} \times \mathcal{Y}$. We define a classifier as a measurable function $f : \mathcal{X} \to \mathbb{R}$. We consider the generalized linear hypothesis class

$$\mathcal{H}_g = \{f(x) = g(w \cdot x) + b : |w| = 1, |b| \leq G\}, \tag{11}$$

where $g : \mathbb{R} \to \mathbb{R}$, and $G > 0$.

Given a rejection threshold $\rho \in (0, \frac{1}{2})$ and a rejection cost $c \in (0, \frac{1}{2})$, the rejection loss is defined by

$$\ell_\rho(f, x, y) = (1 - c)\,\mathbf{1}_{\{yf(x) < -\rho\}} + c\,\mathbf{1}_{\{yf(x) \leq \rho\}}.$$

We assume that the instance space $\mathcal{X}$ is the $L_2$ unit ball $\mathcal{X} = B_2(0, 1)$. For $x \in \mathcal{X}$, the adversarial neighborhood is

$$B_2(x, \gamma) = \{x' \in \mathcal{X} : \|x' - x\|_2 \leq \gamma\},$$

where $\gamma > 0$ is the adversarial budget. The corresponding adversarial reject loss is

$$\ell_\rho^\gamma(f, x, y) = \sup_{\|x' - x\| \leq \gamma} \ell_\rho(f, x', y) = \varphi\left(\inf_{\|x' - x\| \leq \gamma} yf(x')\right), \tag{12}$$

where $\varphi(t) = (1 - c)\mathbf{1}_{\{t < -\rho\}} + c\,\mathbf{1}_{\{t \leq \rho\}}$.

For $\eta(x) = P(Y = 1 \mid X = x)$ and any loss $\ell$, the conditional risk is

$$C_\ell(f, x, \eta) = \eta\,\ell(f, x, 1) + (1 - \eta)\,\ell(f, x, -1),$$

and the minimal (respectively pseudo-minimal) conditional risk over $\mathcal{H}_g$ is

$$C^\star_{\mathcal{H}_g, \ell}(x, \eta) = \inf_{f \in \mathcal{H}_g} C_\ell(f, x, \eta) \quad (\text{resp } C^\star_{\mathcal{H}_g, \ell}(\eta) = \inf_{f \in \mathcal{H}_g,\, x \in \mathcal{X}} C_\ell(f, x, \eta)).$$

Unless otherwise stated, surrogate losses are margin-based and take the form

$$\ell(f, x, y) = \phi(yf(x)), \qquad \phi : \mathbb{R} \to \mathbb{R}_{\geq 0}.$$

Let us define $\underline{m} = \max_{\alpha \in \mathcal{A}_{\mathcal{F}_1}} (g(\alpha) - g(\alpha - \gamma))$ and $\overline{m} = \min_{\alpha \in \mathcal{A}_{\mathcal{F}_1}} (g(\alpha) - g(\alpha + \gamma))$.

Let us assume that $g$ is non decreasing. Hence, as shown in [13], the loss (12) can be rewritten as:

$$\ell_\rho^\gamma(f, x, y) = \varphi\left(\inf_{\|x' - x\| \leq \gamma} yf(x')\right)$$
$$= (1 - c)\mathbf{1}_{\{yg(w.x - \gamma y) + by < -\rho\}} + c\,\mathbf{1}_{\{yg(w.x - \gamma y) + by \leq \rho\}} \tag{13}$$

Let $\mathcal{F}_1$, $\mathcal{F}_2$ be defined as

$$\mathcal{F}_1 = \{\, x \mapsto w \cdot x \mid \|w\| = 1 \,\} \quad \text{and} \quad \mathcal{F}_2 = \{\, x \mapsto b \mid |b| \leq G \,\}.$$

Note that for any $f \in \mathcal{H}_g$ and $x \in \mathcal{X}$, there exist $\alpha_1 \in \mathcal{A}_{\mathcal{F}_1}$ and $\alpha_2 \in \mathcal{A}_{\mathcal{F}_2}$ such that

$$f(x) = g(\alpha_1) + \alpha_2,$$

where

$$\mathcal{A}_{\mathcal{F}_1} \stackrel{\text{def}}{=} \{f_1(x) \mid f_1 \in \mathcal{F}_1,\ x \in \mathcal{X}\}, \qquad \mathcal{A}_{\mathcal{F}_2} \stackrel{\text{def}}{=} \{f_2(x) \mid f_2 \in \mathcal{F}_2,\ x \in \mathcal{X}\}.$$

Therefore, the adversarial reject-option loss $\ell_\rho^\gamma(f, x, y)$ can be equivalently expressed in terms of $(\alpha_1, \alpha_2)$ as

$$\ell_\rho^\gamma(\alpha_1, \alpha_2, y) = (1-c)\mathbf{1}_{\{yg(\alpha_1 - \gamma y) + \alpha_2 y < -\rho\}} + c\,\mathbf{1}_{\{yg(\alpha_1 - \gamma y) + \alpha_2 y \leq \rho\}} \tag{14}$$

We can therefore rewrite the conditional risk as:

$$C_{\ell_\rho^\gamma}(\alpha_1, \alpha_2, \eta) = \eta\,\ell_\rho^\gamma(\alpha_1, \alpha_2, 1) + (1-\eta)\,\ell_\rho^\gamma(\alpha_1, \alpha_2, -1), \tag{15}$$

and equivalently, for a surrogate margin-based loss $\phi$

$$C_\phi(\alpha_1, \alpha_2, \eta) = \eta\,\phi(g(\alpha_1) + \alpha_2) + (1-\eta)\,\phi(-g(\alpha_1) - \alpha_2), \tag{16}$$

$$C^\star_{\mathcal{H}_g, \ell_\rho^\gamma}(\eta) = \inf_{\alpha_1 \in \mathcal{A}_{\mathcal{F}_1},\, \alpha_2 \in \mathcal{A}_{\mathcal{F}_2}} C_{\ell_\rho^\gamma}(\alpha_1, \alpha_2, \eta), \quad C^\star_{\mathcal{H}_g, \phi}(\eta) = \inf_{\alpha_1 \in \mathcal{A}_{\mathcal{F}_1},\, \alpha_2 \in \mathcal{A}_{\mathcal{F}_2}} C_\phi(\alpha_1, \alpha_2, \eta).$$

We also define the excess conditional risks as:

$$\Delta C_{\ell_\rho^\gamma}(\alpha_1, \alpha_2, \eta) = C_{\ell_\rho^\gamma}(\alpha_1, \alpha_2, \eta) - C^\star_{\mathcal{H}_g, \ell_\rho^\gamma}(\eta), \quad \Delta C_\phi(\alpha_1, \alpha_2, \eta) = C_\phi(\alpha_1, \alpha_2, \eta) - C^\star_{\mathcal{H}_g, \phi}(\eta)$$

## A.2  Proof of Lemma 2.6

**Lemma A.1.**
*Let $\mathcal{H}_g$ be the hypothesis set defined in (11). Let us assume that $g : \mathbb{R} \to \mathbb{R}$ is a non-decreasing function with $g(1 + \gamma) < G - \rho$, $g(-1 - \gamma) > \rho - G$ and $\rho > \frac{1}{2}(\underline{m} - \overline{m})$. Then, if a loss $\ell_2$ is $\mathcal{H}_g$−uniformly calibrated with respect to $\ell_\rho^\gamma$, it is also $\mathcal{H}_g$−pseudo uniformly calibrated with respect to $\ell_\rho^\gamma$.*

*Proof.* By combining (14) and (15), we have

$$C_{\ell_\rho^\gamma}(\alpha_1, \alpha_2, \eta) = \eta\Big[(1-c)\,\mathbf{1}_{\{g(\alpha_1 - \gamma) + \alpha_2 < -\rho\}} + c\,\mathbf{1}_{\{g(\alpha_1 - \gamma) + \alpha_2 \leq \rho\}}\Big]$$
$$+ (1-\eta)\Big[(1-c)\,\mathbf{1}_{\{-g(\alpha_1 + \gamma) - \alpha_2 < -\rho\}} + c\,\mathbf{1}_{\{-g(\alpha_1 + \gamma) - \alpha_2 \leq \rho\}}\Big] \tag{17}$$

Because $g$ is non-decreasing, we can identify six possible cases:

$$C_{\ell_\rho^\gamma}(\alpha_1, \alpha_2, \eta) = \begin{cases} 1 & \text{if } g(\alpha_1 - \gamma) + \alpha_2 < -\rho \text{ and } g(\alpha_1 + \gamma) + \alpha_2 > \rho \quad \text{(C1)}, \\[4pt] \eta & \text{if } g(\alpha_1 + \gamma) + \alpha_2 < -\rho \quad \text{(C2)}, \\[4pt] 1 - \eta & \text{if } g(\alpha_1 - \gamma) + \alpha_2 > \rho \quad \text{(C3)}, \\[4pt] \eta c + (1-\eta) & \text{if } -\rho < g(\alpha_1 - \gamma) + \alpha_2 \leq \rho \text{ and } g(\alpha_1 + \gamma) + \alpha_2 > \rho \quad \text{(C4)}, \\[4pt] \eta + (1-\eta)c & \text{if } g(\alpha_1 - \gamma) + \alpha_2 < -\rho \text{ and } -\rho \leq g(\alpha_1 + \gamma) + \alpha_2 \leq \rho \quad \text{(C5)}, \\[4pt] c & \text{if } -\rho \leq g(\alpha_1 - \gamma) + \alpha_2 \text{ and } g(\alpha_1 + \gamma) + \alpha_2 \leq \rho \quad \text{(C6)}. \end{cases} \tag{18}$$

From (18) it follows that for all $(\alpha_1, \alpha_2)$, $C_{\ell_\rho^\gamma}(\alpha_1, \alpha_2, \eta) \geq \min(\eta, 1 - \eta, c)$. We now verify that, for each $\alpha_1 \in \mathcal{A}_{\mathcal{F}_1}$ (and thus for all $x \in \mathcal{X}$), there exists $\alpha_2 \in \mathcal{A}_{\mathcal{F}_2}$ (a given classifier by adjusting the biais $b$) with $C_{\ell_\rho^\gamma}(\alpha_1, \alpha_2, \eta) = \min(\eta, 1 - \eta, c)$.

Using the assumptions $\rho - G < g(-1 - \gamma)$ and $g(1 + \gamma) < G - \rho$ and the monotonicity of g, we have for all $\alpha_1 \in \mathcal{A}_{\mathcal{F}_1}$:

$$-G < -\rho + g(-1-\rho) \leq -\rho + g(\alpha_1 - \rho) < \rho + g(\alpha_1 - \gamma) \leq \rho + g(\alpha_1 + \rho) \leq \rho + g(1 + \rho) < G \tag{19}$$

Moreover, the assumption $\rho > \frac{1}{2}(\underline{m} - \overline{m})$ is equivalent to

$$-G < -\rho - g(1 + \gamma) < -\rho - g(\alpha_1 - \gamma) < \rho + \overline{m} - \underline{m} - g(\alpha_1 + \gamma) < \rho - g(\alpha_1 + \gamma) \tag{20}$$

By combining (20) and , we see that constraints (C2), (C3), (C6) are admissible and for all $x \in \mathcal{X}$, $C^\star_{\mathcal{H}_g, \ell_\rho^\gamma}(x, \eta)$ does not depend on $x$, and we therefore have $C^\star_{\mathcal{H}_g, \ell_\rho^\gamma}(x, \eta) = C^\star_{\mathcal{H}_g, \ell_\rho^\gamma}(\eta)$ Hence, every loss $\ell_2$ which is $\mathcal{H}_g$−uniformly calibrated with respect to $\ell_\rho^\gamma$, is also $\mathcal{H}_g$−pseudo uniformly calibrated with respect to $\ell_\rho^\gamma$. $\square$

 **A.3   Proof of Theorem 2.7**

259  **Lemma A.2.** *Given a non-decreasing function $g$ such that $g(1 + \gamma) < G - \rho$, $g(-1 - \gamma) > \rho - G$*
260  *and $\rho > \frac{1}{2}(\underline{m} - \overline{m})$. The conditional excess risk $\Delta C_{\ell_\rho^\gamma}$ satisfies:*

$$
\Delta C_{\ell_\rho^\gamma}(\alpha_1, \alpha_2, \eta) = \begin{cases}
\max(1 - \eta, \eta, 1 - c) & \text{if} \quad (C1) \\[1ex]
(\eta - c)\mathbb{1}_{\min(\eta, 1-\eta) - c \geq 0} + |2\eta - 1|\mathbb{1}_{2\eta - 1 > 0}\mathbb{1}_{\min(\eta, 1-\eta) - c < 0} & \text{if} \quad (C2) \\[1ex]
(1 - \eta - c)\mathbb{1}_{\min(\eta, 1-\eta) - c \geq 0} + |2\eta - 1|\mathbb{1}_{2\eta - 1 < 0}\mathbb{1}_{\min(\eta, 1-\eta) - c < 0} & \text{if} \quad (C3) \\[1ex]
[\eta c\mathbb{1}_{2\eta - 1 > 0} + ((1 - \eta) - \eta(1 - c))\mathbb{1}_{2\eta - 1 < 0}]\,\mathbb{1}_{\min(\eta, 1-\eta) - c < 0} & \\
+ (1 - c)(1 - \eta)\mathbb{1}_{\min(\eta, 1-\eta) - c \geq 0} & \text{if} \quad (C4) \\[1ex]
[(\eta - (1 - \eta)(1 - c))\mathbb{1}_{2\eta - 1 > 0} + (1 - \eta)c\mathbb{1}_{2\eta - 1 < 0}]\,\mathbb{1}_{\min(\eta, 1-\eta) - c < 0} & \\
+ (1 - c)\eta\mathbb{1}_{\min(\eta, 1-\eta) - c \geq 0} & \text{if} \quad (C5) \\[1ex]
[(c - (1 - \eta))\mathbb{1}_{2\eta - 1 > 0} + (c - \eta)\mathbb{1}_{2\eta - 1 < 0}]\,\mathbb{1}_{\min(\eta, 1-\eta) - c < 0} & \text{if} \quad (C6).
\end{cases}
\tag{21}
$$

261  *Where (C1), (C2), (C3), (C4), (C5), (C6) are defined in* (18).

262  *Proof.* The result follows from Lemma A.1 and from the case-by-case definition of the conditional
263  risk $C_{\ell_\rho^\gamma}(\alpha_1, \alpha_2, \eta)$ given in (A.8). By definition, the conditional excess risk is

$$
\Delta C_{\ell_\rho^\gamma}(\alpha_1, \alpha_2, \eta) = C_{\ell_\rho^\gamma}(\alpha_1, \alpha_2, \eta) - C^\star_{\mathcal{H}_g, \ell_\rho^\gamma}(\eta),
$$

264  where $C^\star_{\mathcal{H}_g, \ell_\rho^\gamma}(\eta) = \min(\eta, 1 - \eta, c)$ denotes the optimal conditional risk (Lemma A.1). We illustrate
265  the derivation for the first two cases.

- Case (C1): $g(\alpha_1 - \gamma) + \alpha_2 < -\rho$ and $g(\alpha_1 + \gamma) + \alpha_2 > \rho$, $C_{\ell_\rho^\gamma}(\alpha_1, \alpha_2, \eta) = 1$. Conse-
267  quently,

$$
\Delta C_{\ell_\rho^\gamma}(\alpha_1, \alpha_2, \eta) = 1 - \min(\eta, 1 - \eta, c) = \max(1 - \eta, \eta, 1 - c).
$$

- Case (C2): $g(\alpha_1 + \gamma) + \alpha_2 < -\rho$, we distinguish the possible values of $\min(\eta, 1 - \eta, c)$

269    – If $\eta < c$, we have $\Delta C_{\ell_\rho^\gamma} = 0$.
270    – If $\eta > 1 - c$, we have $\Delta C_{\ell_\rho^\gamma}(\alpha_1, \alpha_2, \eta) = |2\eta - 1|\mathbb{1}_{2\eta - 1 > 0}\mathbb{1}_{\min(\eta, 1-\eta) - c < 0}$.
271    – If $\min(\eta, 1 - \eta) \geq c$, we have $\Delta C_{\ell_\rho^\gamma}(\alpha_1, \alpha_2, \eta) = (\eta - c)\mathbb{1}_{\min(\eta, 1-\eta) - c \geq 0}$

272  The remaining cases (C3)–(C6) can be handled analogously.  □

*Proof.* [of Theorem A.3] We first calculate the calibration function stated in Definition 2.4. We have
$\hat{\delta}(\epsilon) = \inf_{\eta \in [0,1]} \bar{\delta}(\epsilon, \eta)$ where

$$
\bar{\delta}(\epsilon, \eta) = \inf_{\substack{\alpha_1 \in \mathcal{A}_{\mathcal{F}_1} \\ \alpha_2 \in \mathcal{A}_{\mathcal{F}_2}}} \{\Delta C_\phi(\alpha_1, \alpha_2, \eta) \mid \Delta C_{\ell_\rho^\gamma}(\alpha_1, \alpha_2, \eta) \geq \epsilon\}
$$

273  The pseudo calibration function $\hat{\delta}$ satisfies $\hat{\delta}(\epsilon) > 0$ for all $\epsilon > 0$ if on only if $\bar{\delta}(\epsilon, \eta)$ for all $\epsilon > 0$
274  and $\eta \in [0, 1]$.
275  We will consider three cases: $\eta > \frac{1}{2}$, $\eta < \frac{1}{2}$, $\eta = \frac{1}{2}$ along with some subcases when necessary.

1. **Case 1: If $\eta > \frac{1}{2}$** (then $\min(1 - \eta, \eta) = 1 - \eta$)
277    **Subcase I:** $\eta > 1 - c$

We have:

$$\Delta C_{\ell_\rho^\gamma}(\alpha_1, \alpha_2\epsilon, \eta) = \begin{cases} \eta & \text{if (C1)} \\ 2\eta - 1 & \text{if (C2)} \\ 0 & \text{if (C3)} \\ \eta c & \text{if (C4)} \\ \eta - (1-\eta)(1-c) & \text{if (C5)} \\ c - (1-\eta) & \text{if (C6)} \end{cases}$$

The ordering of these values changes at a critical threshold of $\eta = \frac{1}{2-c}$.

If $\eta \geq \frac{1}{2-c}$, we have:

$$\eta \geq \eta - (1-\eta)(1-c) \geq 2\eta - 1 \geq \eta c \geq c - (1-\eta) \geq 0$$

- If $\epsilon > \eta$, then the even $\Delta C_{\ell_\rho^\gamma}$ does not happen and we have $\bar{\delta}(\epsilon, \eta) = \infty$.
- If $\eta \geq \epsilon > \eta - (1-\eta)(1-c)$ then

$$\Delta C_{\ell_\rho^\gamma} \geq \epsilon \Leftrightarrow (C1)$$

- If $\eta - (1-\eta)(1-c) \geq \epsilon > 2\eta - 1$, then

$$\Delta C_{\ell_\rho^\gamma} \geq \epsilon \Leftrightarrow (C1) \text{ or } (C5)$$

- If $2\eta - 1 \geq \epsilon > \eta c$ then

$$\Delta C_{\ell_\rho^\gamma} \geq \epsilon \Leftrightarrow (C1) \text{ or } (C5) \text{ or } (C2).$$

- If $\eta c \geq \epsilon > c - (1-\eta)$ then

$$\Delta C_{\ell_\rho^\gamma} \geq \epsilon \Leftrightarrow (C1) \text{ or } (C5) \text{ or } (C2) \text{ or } (C4).$$

- If $c - (1-\eta) \geq \epsilon$ then

$$\Delta C_{\ell_\rho^\gamma} \geq \epsilon \Leftrightarrow (C1) \text{ or } (C5) \text{ or } (C2) \text{ or } (C4) \text{ or } (C6).$$

Therefore, we have $\bar{\delta}(\epsilon, \eta) > 0$ for all $\epsilon > 0$, $\eta \in (1-c, 1]$ and $\eta \geq \frac{1}{2-c}$, if and only if

$$\begin{cases} \inf\limits_{\substack{\alpha_1 \in \mathcal{A}_{\mathcal{F}_1},\ \alpha_2 \in \mathcal{A}_{\mathcal{F}_2} \\ (C1)}} \Delta C_\phi(\alpha_1, \alpha_2, \eta) > 0 & \text{for } \eta \in (1-c, 1),\ \eta \geq \frac{1}{2-c}, \\ & \quad \text{s.t. } \eta \geq \epsilon > \eta - (1-\eta)(1-c), \\ \inf\limits_{\substack{\alpha_1 \in \mathcal{A}_{\mathcal{F}_1},\ \alpha_2 \in \mathcal{A}_{\mathcal{F}_2} \\ (C1) \text{ or } (C5)}} \Delta C_\phi(\alpha_1, \alpha_2, \eta) > 0 & \text{for } \eta \in (1-c, 1),\ \eta \geq \frac{1}{2-c}, \\ & \quad \text{s.t. } \eta - (1-\eta)(1-c) \geq \epsilon > 2\eta - 1, \\ \inf\limits_{\substack{\alpha_1 \in \mathcal{A}_{\mathcal{F}_1},\ \alpha_2 \in \mathcal{A}_{\mathcal{F}_2} \\ (C1) \text{ or } (C5) \text{ or } (C2)}} \Delta C_\phi(\alpha_1, \alpha_2, \eta) > 0 & \text{for } \eta \in (1-c, 1),\ \eta \geq \frac{1}{2-c}, \\ & \quad \text{s.t. } 2\eta - 1 \geq \epsilon > \eta c, \\ \inf\limits_{\substack{\alpha_1 \in \mathcal{A}_{\mathcal{F}_1},\ \alpha_2 \in \mathcal{A}_{\mathcal{F}_2} \\ (C1) \text{ or } (C5) \text{ or } (C2) \text{ or } (C4)}} \Delta C_\phi(\alpha_1, \alpha_2, \eta) > 0 & \text{for } \eta \in (1-c, 1),\ \eta \geq \frac{1}{2-c}, \\ & \quad \text{s.t. } \eta c \geq \epsilon > c - (1-\eta), \\ \inf\limits_{\substack{\alpha_1 \in \mathcal{A}_{\mathcal{F}_1},\ \alpha_2 \in \mathcal{A}_{\mathcal{F}_2} \\ (C1) \text{ or } (C5) \text{ or } (C2) \text{ or } (C4) \text{ or } (C6)}} \Delta C_\phi(\alpha_1, \alpha_2, \eta) > 0 & \text{for } \eta \in (1-c, 1),\ \eta \geq \frac{1}{2-c}, \\ & \quad \text{s.t. } c - (1-\eta) \geq \epsilon. \end{cases}$$

$$(22)$$

Moreover, we can easily see that:

$$\left\{ \eta \in (1-c, 1),\ \eta \geq \tfrac{1}{2-c} \ \middle|\ \eta \geq \epsilon > \eta - (1-\eta)(1-c) \right\} = \left\{ \eta \in (1-c, 1),\ \eta \geq \tfrac{1}{2-c} \right\}$$

$$\left\{ \eta \in (1-c, 1),\ \eta \geq \tfrac{1}{2-c} \ \middle|\ \eta - (1-\eta)(1-c) \geq \epsilon > 2\eta - 1 \right\} = \left\{ \eta \in (1-c, 1),\ \eta \geq \tfrac{1}{2-c} \right\}$$

$$\left\{ \eta \in (1-c, 1),\ \eta \geq \tfrac{1}{2-c} \ \middle|\ 2\eta - 1 \geq \epsilon > \eta c \right\} = \left\{ \eta \in (1-c, 1),\ \eta \geq \tfrac{1}{2-c} \right\}$$

$$\left\{ \eta \in (1-c,1),\ \eta \geq \tfrac{1}{2-c} \ \middle|\ \eta c \geq \epsilon > c - (1-\eta) \right\} = \left\{ \eta \in (1-c,1),\ \eta \geq \tfrac{1}{2-c} \right\}$$

$$\left\{ \eta \in (1-c,1),\ \eta \geq \tfrac{1}{2-c} \ \middle|\ c - (1-\eta) \geq \epsilon \right\} = \left\{ \eta \in (1-c,1),\ \eta \geq \tfrac{1}{2-c} \right\}$$

Therefore, we reduce the condition (22) to

$$\inf_{\substack{\alpha_1 \in \mathcal{A}_{\mathcal{F}_1},\ \alpha_2 \in \mathcal{A}_{\mathcal{F}_2} \\ (C1)\ \text{or}\ (C5)\ \text{or}\ (C2)\ \text{or}\ (C4)\ \text{or}\ (C6)}} \Delta C_\phi(\alpha_1, \alpha_2, \eta) > 0$$

And

$$\inf_{\substack{\alpha_1 \in \mathcal{A}_{\mathcal{F}_1},\ \alpha_2 \in \mathcal{A}_{\mathcal{F}_2} \\ g(\alpha_1-\gamma)+\alpha_2 \leq \rho}} \Delta C_\phi(\alpha_1, \alpha_2, \eta) > 0$$

since the constraints $(C1), (C2), (C3), (C4), (C5), (C6)$ form a partition of the set $\mathcal{A}_{\mathcal{F}_1} \times \in \mathcal{A}_{\mathcal{F}_2}$. Similarly, the same result fo $\eta < \tfrac{1}{2-c}$ (whenever applicable).

Therefore, when $\eta > 1 - c$, we have $\bar{\delta}(\epsilon, \eta) > 0$ for all $\epsilon > 0$, if and only if

$$\inf_{\substack{\alpha_1 \in \mathcal{A}_{\mathcal{F}_1},\ \alpha_2 \in \mathcal{A}_{\mathcal{F}_2} \\ g(\alpha_1-\gamma)+\alpha_2 \leq \rho}} \Delta C_\phi(\alpha_1, \alpha_2, \eta) > 0$$

**Subcase II:** $\eta \leq 1 - c$

We have:

$$\Delta C_{\ell_\rho^\gamma}(\alpha_1, \alpha_2, \epsilon, \eta) = \begin{cases} \eta & \text{if (C1)} \\ \eta - c & \text{if (C2)} \\ 1 - \eta - c & \text{if (C3)} \\ (1-c)(1-\eta) & \text{if (C4)} \\ \eta(1-c) & \text{if (C5)} \\ 0 & \text{if (C6)} \end{cases}$$

The ordering of these values changes at a critical threshold of $\eta_1 = \tfrac{1}{2-c}$ (whenever it is applicable).

By following a reasoning similar to that in the previous subcase, we obtain that when $\eta \leq 1 - c$, we have $\bar{\delta}(\epsilon, \eta) > 0$ for all $\epsilon > 0$ if and only if

$$\inf_{\substack{\alpha_1 \in \mathcal{A}_{\mathcal{F}_1},\ \alpha_2 \in \mathcal{A}_{\mathcal{F}_2} \\ g(\alpha_1-\gamma)+\alpha_2 < -\rho\ \text{or}\ g(\alpha_1+\gamma)+\alpha_2 > \rho}} \Delta C_\phi(\alpha_1, \alpha_2, \eta) > 0$$

2. **Case 2: If $\eta < \tfrac{1}{2}$ (then $\min(1-\eta, \eta) = \eta$)**

**Subcase I:** $\eta < c$

We have

$$\Delta C_{\ell_\rho^\gamma}(\alpha_1, \alpha_2, \epsilon, \eta) = \begin{cases} 1 - \eta & \text{if (C1)} \\ 0 & \text{if (C2)} \\ 1 - 2\eta & \text{if (C3)} \\ 1 - \eta - \eta(1-c) & \text{if (C4)} \\ c(1-\eta) & \text{if (C5)} \\ c - \eta & \text{if (C6)} \end{cases}$$

The ordering of these values changes at a critical threshold of $\eta_0 = \tfrac{1}{2-c}$. For instance, when $\eta < \tfrac{1-c}{2-c}$, we have

$$1 - \eta > 1 - \eta - \eta(1-c) > 1 - 2\eta > (1-\eta)c > c - \eta$$

and when $\eta > \frac{1-c}{2-c}$,

$$1 - \eta > 1 - \eta - \eta(1 - c) > (1 - \eta)c > 1 - 2\eta > c - \eta$$

By distinguishing between these subcases (whenever applicable) and proceeding similarly to the previous case, we obtain that $\bar{\delta}(\epsilon, \eta) > 0$ for all $\epsilon > 0$ if and only if

$$\inf_{\substack{\alpha_1 \in \mathcal{A}_{\mathcal{F}_1}, \, \alpha_2 \in \mathcal{A}_{\mathcal{F}_2} \\ g(\alpha_1 + \gamma) + \alpha_2 \geq -\rho}} \Delta C_\phi(\alpha_1, \alpha_2, \eta) > 0$$

**Subcase II:** $\eta \geq c$

We have

$$\Delta C_{\ell_\rho^\gamma}(\alpha_1, \alpha_2, \epsilon, \eta) = \begin{cases} 1 - \eta & \text{if (C1)} \\ \eta - c & \text{if (C2)} \\ 1 - \eta - c & \text{if (C3)} \\ (1 - \eta)(1 - c) & \text{if (C4)} \\ \eta(1 - c) & \text{if (C5)} \\ 0 & \text{if (C6)} \end{cases}$$

Once again, the ordering of these values changes at the critical value $\eta_0$, by taking this into account, we obtain that when $\eta \geq c$, we have $\bar{\delta}(\epsilon, \eta) > 0$ for all $\epsilon > 0$ if and only if

$$\inf_{\substack{\alpha_1 \in \mathcal{A}_{\mathcal{F}_1}, \, \alpha_2 \in \mathcal{A}_{\mathcal{F}_2} \\ g(\alpha_1 - \gamma) + \alpha_2 < -\rho \text{ or } g(\alpha_1 + \gamma) + \alpha_2 > \rho}} \Delta C_\phi(\alpha_1, \alpha_2, \eta) > 0$$

3. **Case 3: If $\eta = \frac{1}{2}$**

We have:

$$\Delta C_{\ell_\rho^\gamma}\left(\alpha_1, \alpha_2, \epsilon, \frac{1}{2}\right) = \begin{cases} 1 - c & \text{if (C1)} \\ \frac{1}{2} - c & \text{if (C2) or (C3)} \\ \frac{1}{2}(1 - c) & \text{if (C4) or (C5)} \\ 0 & \text{if (C6)} \end{cases}$$

It follows that $\bar{\delta}(\epsilon, \frac{1}{2}) > 0$ for all $\epsilon > 0$ if and only if

$$\inf_{\substack{\alpha_1 \in \mathcal{A}_{\mathcal{F}_1}, \, \alpha_2 \in \mathcal{A}_{\mathcal{F}_2} \\ g(\alpha_1 - \gamma) + \alpha_2 < -\rho \text{ or } g(\alpha_1 + \gamma) + \alpha_2 > \rho}} \Delta C_\phi\left(\alpha_1, \alpha_2, \frac{1}{2}\right) > 0$$

By noticing that

$$\Delta C_\phi(\alpha_1, \alpha_2, \eta) = C_\phi(\alpha_1, \alpha_2, \eta) - \inf_{\alpha_1 \in \mathcal{A}_{\mathcal{F}_1}, \, \alpha_2 \in \mathcal{A}_{\mathcal{F}_2}} C_\phi(\alpha_1, \alpha_2, \eta)$$

and by gathering all the cases, we conclude that $\phi$ is pseudo-uniform calibrated with respect to $\ell_\rho^\gamma$ if and only if:

$$\inf_{\substack{\alpha_1 \in \mathcal{A}_{\mathcal{F}_1}, \, \alpha_2 \in \mathcal{A}_{\mathcal{F}_2} \\ g(\alpha_1 - \gamma) + \alpha_2 < -\rho \text{ or } g(\alpha_1 + \gamma) + \alpha_2 > \rho}} C_\phi(\alpha_1, \alpha_2, \eta) > \inf_{\alpha_1 \in \mathcal{A}_{\mathcal{F}_1}, \, \alpha_2 \in \mathcal{A}_{\mathcal{F}_2}} C_\phi(\alpha_1, \alpha_2, \eta) \quad \text{if } \min(\eta, 1 - \eta) \geq c$$

$$\inf_{\substack{\alpha_1 \in \mathcal{A}_{\mathcal{F}_1}, \, \alpha_2 \in \mathcal{A}_{\mathcal{F}_2} \\ g(\alpha_1 + \gamma) + \alpha_2 \geq -\rho}} C_\phi(\alpha_1, \alpha_2, \eta) > \inf_{\alpha_1 \in \mathcal{A}_{\mathcal{F}_1}, \, \alpha_2 \in \mathcal{A}_{\mathcal{F}_2}} C_\phi(\alpha_1, \alpha_2, \eta) \quad \text{if } \eta < c$$

$$\inf_{\substack{\alpha_1 \in \mathcal{A}_{\mathcal{F}_1}, \, \alpha_2 \in \mathcal{A}_{\mathcal{F}_2} \\ g(\alpha_1 - \gamma) + \alpha_2 \leq \rho}} C_\phi(\alpha_1, \alpha_2, \eta) > \inf_{\alpha_1 \in \mathcal{A}_{\mathcal{F}_1}, \, \alpha_2 \in \mathcal{A}_{\mathcal{F}_2}} C_\phi(\alpha_1, \alpha_2, \eta) \quad \text{if } \eta > 1 - c$$

Moreover, we have that $\alpha_1 \in [-1, 1]$, $\alpha_2 \in [-G, G]$ for all $\alpha_1 \in \mathcal{A}_{\mathcal{F}_1}$, $\alpha_2 \in \mathcal{A}_{\mathcal{F}_2}$. Therefore, since $g$ is continuous and non-decreasing, it follows that $(g(\alpha_1) + \alpha_2 \in [g(-1) - G, \ g(1) + G]$. hence we have:

$$\{g(\alpha_1) + \alpha_2 : \alpha_1 \in \mathcal{A}_{\mathcal{F}_1}, \ \alpha_2 \in \mathcal{A}_{\mathcal{F}_2}, \alpha_2 \leq \rho - g(\alpha_1 - \gamma)\} = [g(-1) - G, \rho + \underline{m}]$$

$$\{g(\alpha_1) + \alpha_2 : \alpha_1 \in \mathcal{A}_{\mathcal{F}_1}, \ \alpha_2 \in \mathcal{A}_{\mathcal{F}_2}, \alpha_2 \geq -\rho - g(\alpha_1 + \gamma)\} = [\overline{m} - \rho, g(1) + G]$$

$$\{g(\alpha_1) + \alpha_2 : \alpha_1 \in \mathcal{A}_{\mathcal{F}_1}, \ \alpha_2 \in \mathcal{A}_{\mathcal{F}_2}, \alpha_2 < -\rho - g(\alpha_1 - \gamma)\} = [g(-1) - G, -\rho + \underline{m}[$$

$$\{g(\alpha_1) + \alpha_2 : \alpha_1 \in \mathcal{A}_{\mathcal{F}_1}, \ \alpha_2 \in \mathcal{A}_{\mathcal{F}_2}, \alpha_2 > \rho - g(\alpha_1 + \gamma)\} = ]\overline{m} + \rho, g(1) + G]$$

where

$$\underline{m} = \max_{\alpha \in \mathcal{A}_{\mathcal{F}_1}} (g(\alpha) - g(\alpha - \gamma)) = \max_{\alpha \in [-1, 1]} (g(\alpha) - g(\alpha - \gamma))$$

and

$$\overline{m} = \min_{\alpha \in \mathcal{A}_{\mathcal{F}_1}} (g(\alpha) - g(\alpha + \gamma)) = \min_{\alpha \in [-1, 1]} (g(\alpha) - g(\alpha + \gamma)).$$

Additionally, let $\alpha = g(\alpha_1) + \alpha_2$ for all $\alpha_1 \in \mathcal{A}_{\mathcal{F}_1}$, $\alpha_2 \in \mathcal{A}_{\mathcal{F}_2}$, then:

$$C_\phi(\alpha_1, \alpha_2, \eta) = \eta\phi(\alpha) + (1 - \eta)\phi(-\alpha) := \tilde{C}_\phi(\alpha, \eta)$$

The calibration conditions therefore become:

$$\inf_{\alpha \in [g(-1) - G, \ \underline{m} - \rho[ \ \cup \ ]\overline{m} + \rho, \ g(1) + G]} \tilde{C}_\phi(\alpha, \eta) > \inf_{\alpha \in [g(-1) - G, \ g(1) + G]} \tilde{C}_\phi(\alpha, \eta) \quad \text{if } \min(1 - \eta, \eta) \geq c, \quad \text{(rejection)}$$

$$\inf_{\alpha \in [g(-1) - G, \ \underline{m} + \rho]} \tilde{C}_\phi(\alpha, \eta) > \inf_{\alpha \in [g(-1) - G, \ g(1) + G]} \tilde{C}_\phi(\alpha, \eta) \quad \text{if } \eta > 1 - c, \quad \text{(positive classification)}$$

$$\inf_{\alpha \in [\overline{m} - \rho, g(1) + G]} \tilde{C}_\phi(\alpha, \eta) > \inf_{\alpha \in [g(-1) - G, \ g(1) + G]} \tilde{C}_\phi(\alpha, \eta) \quad \text{if } \eta < c, \quad \text{(negative classification)}$$

$\square$

## A.4 Proof of Corollary 2.8

*Proof.* The proof follows immediatly by evaluating $\overline{m}$ and $\underline{m}$ for $g = Relu$, which yields $\overline{m} = -\gamma$ and $\underline{m} = \gamma$. $\square$

## A.5 Proof of Theorem 2.10

*Proof.* The proof relies on a key property of continuous and quasi-concave function. For $\eta = \frac{1}{2}$, the calibration condition is given by:

$$\inf_{\alpha \in [g(-1) - G, \ \underline{m} - \rho[ \ \cup \ ]\overline{m} + \rho, \ g(1) + G]} \tilde{C}_\phi(\alpha, \frac{1}{2}) > \inf_{\alpha \in [g(-1) - G, \ g(1) + G]} \tilde{C}_\phi(\alpha, \frac{1}{2}) \quad (23)$$

Since $\phi$ is continuous and quasi-concave even, it follows from Lemma 32 in [1] that

$$\inf_{\alpha \in [g(-1) - G, \ g(1) + G]} \tilde{C}_\phi(\alpha, \frac{1}{2}) = \min\left(\tilde{C}_\phi\left(g(-1) - G, \frac{1}{2}\right), \tilde{C}_\phi\left(g(1) + G, \frac{1}{2}\right)\right)$$

Moreover, the left-hand side of (23) is necessarily $\min\left(\tilde{C}_\phi\left(g(-1) - G, \frac{1}{2}\right), \tilde{C}_\phi\left(g(1) + G, \frac{1}{2}\right)\right)$ which leads to a contradiction. $\square$

## A.6 Possible Calibrated Surrogate losses for $\ell_\rho^\gamma$

We consider the Double Sigmoid Loss (DSL) introduced by [8] defined as:

$$\phi_{DS,\rho}^\mu(yf(x)) = 2c\sigma(yf(x) - \rho) + 2(1-c)\sigma(yf(x) + \rho) \tag{24}$$

with $\sigma(x) = \frac{1}{1+\exp(\mu x)}, \mu > 0$.

We further define a shifted version as

$$\phi_{DS,\rho}^{\mu,\beta} = \phi_{DS,\rho-\beta}^\mu \quad \text{with } \rho > \beta > 0 \tag{25}$$

A careful analysis of previous works[10] reveals that the surrogate losses proposed therein do not adequately capture the rejection regime. This is a significant drawback, as these losses fail to indicate when rejection should be the optimal decision. In contrast, our approach introduces surrogate losses that explicitly incorporate the rejection option. To illustrate this we consider $g = \text{RELU}$, the Shifted Double Sigmoid loss (25) with parameters $c = 0.4, \rho = 0.3, \mu = 30, \gamma = 0.04, \beta = 0.05, G = 4$. The figure 1 shows the minimizers indeed lie within the set prescribed by the calibration condition when rejection is optimal $(\min(1 - \eta, \eta) \geq c)$. Additionnaly, Figure 2 highlights the regime where positive prediction is optimal $(\eta > 1 - c)$, while Figure 3 depicts the corresponding regime for negative prediction $(\eta < c)$.

Guided by these illustrations, we are going to investigated from a theoretical point of view how the parameters $c, \rho, \mu, \gamma, \beta$ interact to guarantee that the Shifted Double Sigmoid Loss is $\mathcal{H}_g-$ calibrated. Establishing such a relation will provide intuition that can be extended to more general function $g$ and to broader classes of surrogate losses.

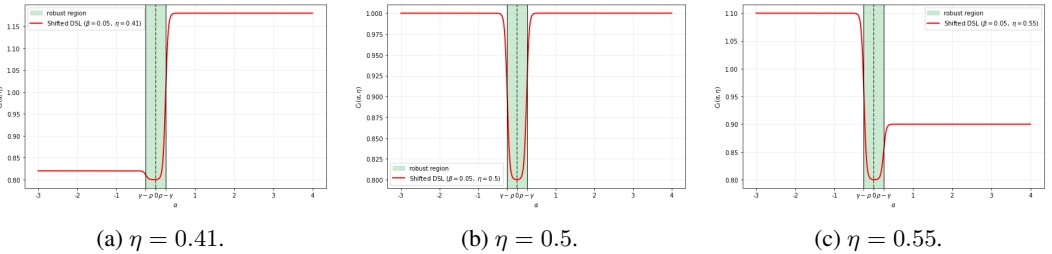

(a) $\eta = 0.41$.      (b) $\eta = 0.5$.      (c) $\eta = 0.55$.

Figure 1: Illustration of minimizers under rejection regime $(\min(\eta, 1 - \eta) \geq c)$.

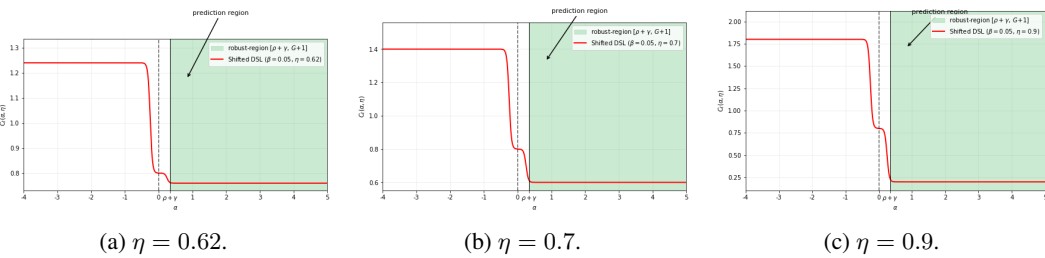

(a) $\eta = 0.62$.      (b) $\eta = 0.7$.      (c) $\eta = 0.9$.

Figure 2: Illustration of minimizers under positive prediction $(\eta > 1 - c)$.

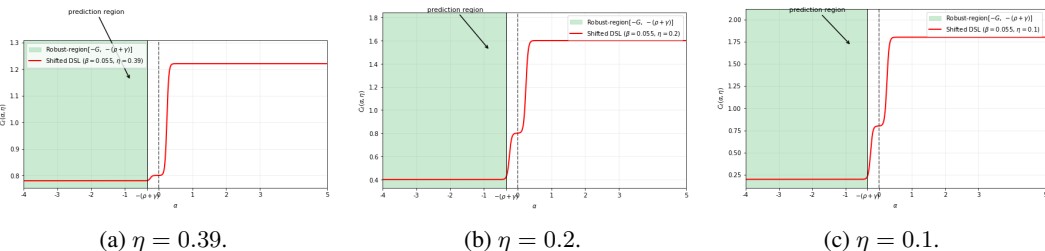

(a) $\eta = 0.39$.      (b) $\eta = 0.2$.      (c) $\eta = 0.1$.

Figure 3: Illustration of minimizers under negative prediction $(\eta < c)$.

