# OpenReview forum: "Calibrated surrogate losses for robust classification with a reject option"
_EurIPS.cc/2025/Workshop/UPLB — UPLB2025_

### Official Review · Reviewer_yyPb · 2025-10-24
**Extension of adversarial calibration theory to include a rejection option, preliminary but aligned with workshop theme**

**Rating:** 6
**Confidence:** 3

**Review:**

The paper extends prior analyses of surrogate-loss calibration for adversarial classification to the setting where the classifier can abstain from prediction when uncertainty exceeds a given threshold (the reject option). The analysis focuses on generalized linear classifiers. The main result shows that while quasi-concave margin-based losses are calibrated in standard adversarial classification, they lose this property once the reject option is introduced, motivating the exploration of non–quasi-concave losses as potential calibrated surrogates.

The work looks quite preliminary at this stage. For instance, it would benefit from even a simple empirical investigation of the non–quasi-concave alternatives. There must have been an issue with the pdf upload as the title is missing and currently displays NeurIPS formatting instructions.

Overall, I would lean toward a mild acceptance. Although the work is limited at this stage, I think it fits the theme of the workshop as it addresses theoretical guarantees for robustness.  I must say I am not an expert in this specific subfield and the mathematical derivations were not checked in detail.

---

### Decision · Program_Chairs · 2025-11-03

Accept (Poster)